# Screening Repurposed Antiviral Small Molecules as Antimycobacterial Compounds by a Lux-Based phoP Promoter-Reporter Platform

**DOI:** 10.3390/antibiotics11030369

**Published:** 2022-03-09

**Authors:** Li Zhu, Annie Wing-Tung Lee, Kelvin Ka-Lok Wu, Peng Gao, Kingsley King-Gee Tam, Rahim Rajwani, Galata Chala Chaburte, Timothy Ting-Leung Ng, Chloe Toi-Mei Chan, Hiu Yin Lao, Wing Cheong Yam, Richard Yi-Tsun Kao, Gilman Kit Hang Siu

**Affiliations:** 1Department of Health Technology and Informatics, The Hong Kong Polytechnic University, Hong Kong, China; li.li.zhu@connect.polyu.hk (L.Z.); wtunglee@polyu.edu.hk (A.W.-T.L.); kelvin-ka-lok.wu@polyu.edu.hk (K.K.-L.W.); rahim.rajwani@nih.gov (R.R.); chala.galata@connect.polyu.hk (G.C.C.); tl-timothy.ng@connect.polyu.hk (T.T.-L.N.); tm-chloe.chan@polyu.edu.hk (C.T.-M.C.); hiu-yin.lao@connect.polyu.hk (H.Y.L.); 2Department of Microbiology, Li Ka Shing Faculty of Medicine, The University of Hong, Hong Kong, China; gaop0305@gmail.com (P.G.); kingskgt@connect.hku.hk (K.K.-G.T.); wcyam@hku.hk (W.C.Y.); rytkao@hku.hk (R.Y.-T.K.)

**Keywords:** *lux*-based promoter-reporter platforms, small-molecule compounds, *Mycobacterium tuberculosis* complex, anti-virulence agents, antibiotics

## Abstract

The emergence of multidrug-resistant strains and hyper-virulent strains of *Mycobacterium tuberculosis* are big therapeutic challenges for tuberculosis (TB) control. Repurposing bioactive small-molecule compounds has recently become a new therapeutic approach against TB. This study aimed to identify novel anti-TB agents from a library of small-molecule compounds via a rapid screening system. A total of 320 small-molecule compounds were used to screen for their ability to suppress the expression of a key virulence gene, *phop*, of the *M. tuberculosis* complex using luminescence (*lux*)-based promoter-reporter platforms. The minimum inhibitory and bactericidal concentrations on drug-resistant *M. tuberculosis* and cytotoxicity to human macrophages were determined. RNA sequencing (RNA-seq) was conducted to determine the drug mechanisms of the selected compounds as novel antibiotics or anti-virulent agents against the *M. tuberculosis* complex. The results showed that six compounds displayed bactericidal activity against *M. bovis* BCG, of which Ebselen demonstrated the lowest cytotoxicity to macrophages and was considered as a potential antibiotic for TB. Another ten compounds did not inhibit the in vitro growth of the *M. tuberculosis* complex and six of them downregulated the expression of phoP/R significantly. Of these, ST-193 and ST-193 (hydrochloride) showed low cytotoxicity and were suggested to be potential anti-virulence agents for *M. tuberculosis*.

## 1. Introduction

Tuberculosis (TB) remains a significant global health problem today. According to the World Health Organization Global TB report in 2021, TB ranked 13th among the leading causes of death worldwide [1]. Approximately 10 million people developed TB, which claimed over 1.5 million lives in 2020 [1]. Approximately 10 million people developed TB, which claimed over 1.4 million lives in 2019 [1]. TB infections associated with multidrug-resistant (MDR), extensively drug-resistant (XDR), and totally drug-resistant (TDR) *Mycobacterium tuberculosis* strains have been increasing in recent years [1,2,3,4]. This situation poses a significant threat to global TB control, particularly in resource-poor countries with a high prevalence of AIDS. However, according to the global new TB drug development pipeline built by the Stop TB Partnership Working Group on New Drugs in 2021, only a few anti-TB drugs were in phase III clinical trials [5], and some of them could not penetrate through complex lung lesions and the MTB cell wall, thereby failing to eliminate *M. tuberculosis* [6,7]. Hence, there is an urgent need to develop new anti-TB treatments against *M. tuberculosis* strains with various patterns of drug resistance.

Drug repurposing is a novel strategy to treat TB, which uncovers known compounds that have the unexpected potential to treat TB [8]. Repurposed drugs such as antibiotics, antifungal, antiviral, and anticancer drugs were potential anti-TB drugs; however, their effectiveness and mechanisms have not been clearly understood [9]. Of them, repurposed antiviral drugs were the least frequently studied. Only a few antiviral drugs, including Isoprinosine, were identified [9,10]. Nevertheless, anti-virulence therapy is an innovative therapeutic strategy to treat multidrug-resistant organisms. It focuses on disarming bacterial virulence factors that facilitate disease development instead of killing the bacteria [11] and often penetrates through the host cell members to eliminate MTB. A previous study discovered a small-molecule compound, 2-phospho-L-ascorbic acid (2P-AC), could reduce mycobacterial survival in macrophage infections [12]. It underlined the potential for the development of anti-virulence agents against *M. tuberculosis*.

Several thousands of small-molecule compounds are approved and passed phase I clinical drug trials, meaning that they have completed extensive preclinical and clinical studies and have well-characterized bioactivities, safety, and bioavailability properties. These compounds could be potentially repurposed to inhibit *M. tuberculosis* virulence. As such, a rapid and high-throughput platform that can screen effective anti-virulence agents for TB is warranted. *PhoP*, which is a key virulence gene of TB, is a global transcriptional regulator of lipid metabolism and hypoxic response and controls the expression of ~2% of the genes in *M. tuberculosis* [13]. It was shown that disruption of *phop* in *M. tuberculosis* caused impaired multiplication within macrophages, suggesting that this gene possibly plays an essential role in the intracellular growth of *M. tuberculosis* [14]. In addition, our previous study identified a common mutation in the promoter region of *phop*, which could confer aggressive intramacrophage growth of hypervirulent *M. tuberculosis* strains [15]. Therefore, we hypothesized that suppression of *phoP* expression might be able to impair intramacrophage survival of *M. tuberculosis*, facilitating the host’s immune system to eradicate the bacteria.

The present study aimed to screen small-molecule compounds that were likely to suppress the expression of *phop* in *Mycobacterium bovis* BCG using a *lux*-based promoter-reporter platform. Compared with *M. tuberculosis*, *M. bovis* BCG has a lower biosafety level (risk group 2 and biosafety level 2 practices), despite its genome being >99.95% identical to that of *M. tuberculosis* reference strain H37Rv [16]. This allowed the screening process to be conducted in the BSL-2 laboratory. A library of 320 small-molecule compounds with antiviral activities was screened for their anti-TB potency. The compounds that could reduce the *lux* signal were selected for further validation on (i) their abilities to inhibit in vitro growth of the *M. tuberculosis* complex, (ii) cytotoxicity to THP-1 macrophage, and (iii) dysregulated expression of *phop* and its associated gene networks. Our findings eventually suggested one potential antibiotic and two anti-virulence agents for anti-TB therapy.

## 2. Results

### 2.1. Construction and Validation of the Lux-Based Promoter-Reporter Platform

The mycobacterial reporter plasmid -pMV306G13+Lux and the *lux*-based *phoP* promoter-reporter plasmid (pMV306PhoP+Lux) were constructed as shown in Figure 1A,B. To validate the platform, ethoxzolamide (ETZ), which could downregulate *phop* expression, was used to demonstrate if the *lux* signal could represent *phop* expression in *M. bovis* BCG. We firstly showed that there was no significant difference (*p*  >  0.05) in OD_600_ between *M. bovis* BCG with and without ETZ treatment (Figure 1(Ci)). This indicated that ETZ did not affect bacterial growth. As we expected, the *lux* signal in *M. bovis* BCG after 24 h of ETZ treatment declined prominently (*p*  <  0.001 vs. the control group without treatment at 24 h, Figure 1(Cii)). Our results also demonstrated a statistical effect size of 0.886, indicating a high quality of screening assay. Meanwhile, RT-qPCR revealed that the expression of the *luxA* and *phoP* genes was significantly suppressed (*p*  <  0.001) after ETZ treatment (Figure 1D). Taken together, the result highlighted that *phop* expression in *M. bovis* BCG after drug treatment could be illustrated by referring to its *lux* signal.

### 2.2. Screening of 320 Antiviral Compounds and 3 Anti-TB Drugs

*M. bovis* BCG transformed with pMV306Adaptor+Lux (control set) and pMV306PhoP+Lux (phoP set) were, respectively, treated with a total of 320 antiviral compounds and 3 anti-TB drugs. For the control set, the *lux* signals were consistently below 50 CPS (no difference existed compared with the blank wells, *p* > 0.05) after all compound treatments, except for triciribine (CPS = 155) (Figure 2A). For the PhoP set, the drug-free controls (Figure 2B, green dots) showed an average *lux* signal at around 300 CPS, which is regarded as the basal *lux* intensity induced by the *phop* promoter in the absence of compound challenges. When *M. bovis* BCG was treated with the three anti-TB drugs (Figure 2B, red dots), no significant change of *lux* signals was observed. This indicated that these three common anti-TB drugs could not repress phoP. Notably, 16 antiviral small compounds (CPS ≈ 0, blue dots in Figure 2B) presented a significant suppression effect on *lux* signals after a 4 h treatment when compared with the drug-free control (Figure 2C). These 16 compounds, which are listed in Appendix A, were selected for further experimental validation. 

### 2.3. MICs and MBCs of Six Compounds against BCG/M. Tuberculosis

After 4 h of treatment with the above 16 compounds, *M. bovis* BCG were washed and subcultivated in drug-free media, followed by incubation for 14 days. No viable *M. bovis* BCG was observed after treatment with six compounds, namely Trifluoperazine (dihydrochloride), Elvitegravir, NH125, Ebselen, Letrazuril, and Shikonin, whereas confluent growth was observed after treatment with the remaining 10 compounds. Subsequently, the MICs and MBCs of the six compounds against *M. bovis* BCG, *M. tuberculosis* H37Rv, and the two drug-resistant clinical isolates of *M. tuberculosis*, HKU14621 (MDR-TB) and WC274 (XDR-TB), were determined. Except for Elvitegravir and Letrazuril, these compounds demonstrated bactericidal effects on *M. bovis* BCG as well as all *M. tuberculosis* strains at a concentration of 100 μM or below (Table 1). Notably, the MICs and MBCs against the MDR-TB and XDR-TB strains were almost identical to those against *M. tuberculosis* H37Rv, suggesting that these compounds can kill MDR- and XDR-*M. tuberculosis* clinical isolates as effective as the pan-susceptible reference strain (Table 1).

### 2.4. RNA-Seq Transcriptome Analysis

To unveil the genetic mechanisms of drug actions of the 16 selected compounds on the *M. tuberculosis* complex, the transcriptomes of *M. bovis* BCG after compound treatment were profiled using RNA-seq. Based on the genome-wide differential expression patterns, the datasets could be clustered into two groups by a principal component analysis (PCA) and hierarchical clustering (Figure 3A,B). The clustering separated the compounds with bactericidal activity from those without an inhibitory effect. The compound group in which *M. bovis* BCG “survived” after the treatment was labelled as the survival (S) group, whereas the compounds that led to the “death” of *M. bovis* BCG were labelled as the dead (D) group. Interestingly, four compounds that belonged to the S group, namely Pirodavir, AEBSF (hydrochloride), Bicyclol, and ABX464, exhibited similar transcriptome profiles and merged into the same cluster as the D group (Figure 3B). Hence, we labelled these four compounds as S2 while the other six compounds were named S1.

### 2.5. Expression of phoP-Associated Pathways upon Compound Treatment

*PhoP* was differentially expressed between S1 and the other groups (two-way ANOVA, *p*  <  0.05 vs. S2 and Control and *p*  <  0.001 vs. D), while *phoR* was significantly downregulated in S1 when compared with D (two-way ANOVA, *p*  <  0.05) (Figure 3C). This indicated that the compounds in S1 could effectively downregulate *phoP* expression in *M. bovis* BCG. Next, the expression of *phoP* and its downstream gene network was investigated specifically. Interestingly, in addition to *phop*, the related genes in the associated pathways were consistently downregulated when *M. bovis* BCG was treated with two compounds, ST-193 (B4) and ST-193 (hydrochloride) (B6) (Figure 3D). This implicated that these two compounds might dysregulate the entire *phoP*-associated pathways.

### 2.6. Molecular Regulation Associated with the Anti-Virulence Process

To identify differentially expressed genes (DEGs) between the S and D groups, genes with at least a 2-fold difference and an adjusted *p*  < 0.05 were selected. Altogether, 52 DEGs (27 upregulated and 25-downregulated) were identified in groups D vs. S (Figure 4A, Appendix A). Moreover, for the DEGs between the D and S2 groups, only one gene, *phoY1*, was identified (Figure 4B, Appendix A). Next, a gene ontology (GO) enrichment analysis was performed to reveal the important biological processes and molecular functions dysregulated between the S and D groups. We demonstrated that genes involved in multidrug resistance mechanisms (eff*lux* pump), DNA repair system, PPE family protein, and polyketide synthesis were downregulated in the dead group, while genes related to the DNA repair process and cell wall biogenesis were downregulated in the survival (S) group (Figure 4C, Appendix A). The results highlighted the importance of these processes during anti-virulence.

### 2.7. Prediction of Drug Targets in the Dead Group

As described above, six compounds exhibited bactericidal activity against *M. bovis* BCG. The DEG profiles were investigated to identify the potential drug targets of these compounds. Some common downregulated genes (logFC < −2 and FDR < 0.1 when compared with the DMSO control sample), including *cysA2*, *frdC*, and *glpD2_2*, were identified when *M. bovis* BCG was treated with Ebselen (A3), NH125 (A9), and Shikonin (B2) (Appendix A). It should be noted that the sequences of *cysA2*, *frdC*, and *glpD2_2* in *M. bovis* BCG were almost 100% identical to those in *M. tuberculosis* H37Rv. It is consistent with our observation in phenotypic drug susceptibility tests that Ebselen, NH125, and Shikonin shared very similar MICs and MBCs against the *M. bovis* BCG and *M. tuberculosis* strains. Therefore, *cysA2*, *frdC*, and *glpD2* were considered as potential drug targets of these small compounds.

### 2.8. THP-1 Cytotoxicity of the Small-Molecule Compounds

The cytotoxicity of the 16 selected small compounds was determined using a THP-1 macrophage. Overall, five compounds were considered as non-cytotoxic to human macrophages given that their CC_50_ were greater than 200 μM, the highest achievable concentration in this study (Figure 5, Appendix A). Of these five compounds, ST-193 and ST-193 (hydrochloride) were shown to dysregulate the expression of most genes in the *phop*-associated pathways without inhibiting the in vitro growth of *M. bovis* BCG, suggesting that they were potential anti-virulence agents in anti-TB therapy. Moreover, among the compounds that had a bactericidal effect on *M. tuberculosis*, only Ebselen had the CC_50_ (>200 μM) greater than its MIC (50 μM) and MBC (100 μM) against the *M. tuberculosis* complex. It was, therefore, considered as a potential antibiotic for *M. tuberculosis*.

## 3. Discussion

Repurposing bioactive small-molecule compounds has been suggested as a new therapeutic approach against *M. tuberculosis* infection. Unlike previous studies, which focused on one or several compounds [12], we examined the anti-TB potency of a library of 320 small-molecule compounds with antiviral activity. As antiviral agents, these compounds are expected to effectively penetrate through the host cell members, which is an important feature of the drugs used to treat intracellular pathogens, such as the *M. tuberculosis* complex. In addition to identifying new candidates of antibiotic from these compounds, this study aimed to discover anti-virulence agents for *M. tuberculosis* that impaired the bacterial virulence factors without killing the organisms [11]. Notably, the development of anti-virulence drugs requires an in-depth understanding of the roles that diverse virulence factors have in disease processes. Previous studies from our team and other research groups suggested that *phop* is essential for the intracellular growth of the *M. tuberculosis* complex inside macrophages [14,15]. Disruption of *phop* could impair intramacrophage and facilitate the host’s immune system to eradicate the bacteria.

Instead of individual measurement of *phop* expression using RT-qPCR, we developed a *lux*-based promoter-reporter screening platform to select potential agents that can inhibit *phoP* promoter activity. This platform enabled the real-time measurement of *phop* expression in viable *M. bovis BCG* culture in response to the compound challenge, in terms of *lux* signal, in a batch of 96 samples. In the screening platform validation, we successfully demonstrated that ETZ could reduce the RNA quantity of both *phop* and *lux* signals, indicating the luminescence generated by the phoP promoter-reporter plasmid, pMV306PhoP+Lux, corresponded to the *phop* expression in *M. BCG* bovis.

By using our screening platform, 16 out of 320 (5.0%) small-molecule compounds were identified as potential candidates against *M. tuberculosis* infection. Surprisingly, six of them: trifluoperazine (dihydrochloride), elvitegravir, NH125, ebselen, letrazuril, and shikonin did not show a decrease in *phoP* gene expression and directly eliminated *M. bovis* BCG. The decrease in their *lux* signals was possibly caused by bacterial death. In the molecular regulation analysis, we highlighted that dysregulation of the multidrug resistance mechanisms, DNA repair system, PPE family protein, and polyketide synthesis would determine *M. tuberculosis* death. Among the six compounds, only Ebselen exhibited a CC_50_ greater than its MIC and MBC against the *M. tuberculosis* complex, including the H37Rv, MDR-MTB and XDR-MTB strains. Our RNA-seq analysis identified that Ebselen displayed bactericidal activity in *M. bovis* BCG through *cysA2* and *glpD2* downregulation, which are the major regulators of sulfur and glycerol metabolism, respectively. *CysA2* is an essential regulator in the sulfur assimilation pathway [17], and its dysregulation could impair *M. tuberculosis* survival in macrophages [18]. In parallel, a previous study demonstrated that the inhibition of glycerol metabolism by 2-aminoquinazolinones via *glpD2* downregulation could kill *M. tuberculosis* in vitro [19]. Taken together, dysfunctions of sulfur and glycerol metabolism are possible drug mechanisms for Ebselen. It might also be effective on *M. tuberculosis,* as its sequences of *cysA2* and *glpD2* were almost 100% identical to those in *M. bovis* BCG. Hence, Ebselen could be considered as a potential antibiotic for the *M. tuberculosis* complex. In addition, it is interesting to highlight that *phoY1*, which controls phosphate sensing in *M. tuberculosis*, thereby affecting its susceptibility to antibiotics [20], was the only downregulated DEG between the dead group and the survival groups. This might indicate the importance of *phoY1* on *M. tuberculosis* survival.

Among the remaining 10 survival candidates, 6 of them, Saquinavir, Clemizole (hydrochloride), Bay 41-4109 (racemate), ST-193, ST-193 (hydrochloride) and Saquinavir (Mesylate), exhibited a distinct transcriptome profile and a lower gene expression of *phoP*. In the molecular regulation analysis, genes related to the DNA repair process and cell wall biogenesis in this group were significantly downregulated. It was found that the activity of the PhoP-related system interacted with genes involved in the fadD family, which regulates the fatty acid β-oxidative pathway [21]. Our results revealed that genes in the fadD family (*fadD29* and *fadD22*) are of lower expression in *M. bovis* BCG after being treated with these six compounds. Disruption of fatty acid biosynthesis could inhibit bacterial differentiation and growth [22]. These six compounds are potential anti-virulence agents because they inactivated *M. bovis* BCG through DNA damage, cell wall destruction, and differentiation inhibition, thereby facilitating the host’s immune system to eradicate the bacteria. Interestingly, ST-193 and ST-193 (hydrochloride), with CC_50_ greater than 200 μM, dysregulated most genes in the *phoP*-associated network in *M. bovis* BCG, so they are the most desirable anti-virulence agents. In brief, many studies revealed that ST-193 effectively treats guinea pig models of Lassa virus infections [23,24,25,26]. ST-193 was found to suppress the conformational rearrangement of the arenavirus envelope glycoprotein, which is necessary for membrane fusion [27]. However, the drugs possibly affect more targets that we had not uncovered; therefore, the mechanism and effectiveness of ST-193 and ST-193 (hydrochloride) against *M. tuberculosis* warrant further investigation.

For proof-of-concept, only 320 small compounds were screened in this study. Despite this small-scale screening, three (0.94%) compounds were identified as potential antibiotic and anti-virulence agents *against M. tuberculosis*. In the future, the discovery of more new candidates of anti-TB drugs is anticipated if more drug repurposing compounds are screened using the promoter-reporter platforms coupled with automatic high-throughput screening instruments. Current TB treatment relies on a synergistic combination of drugs administered for the desired time to ensure definitive non-relapsing cures and to avoid drug-resistant mutants [28]. In 2021, the World Health Organization (WHO) established a “*Position statement on innovative clinical trial design for development of new TB treatments*” to outline clinical trial designs [29]. Anti-virulence agents facilitate the development of new TB therapies and provide us an avenue to establish our innovative TB treatment in clinical trials. However, it is necessary to see if the anti-virulence agents have synergistic or antagonistic interactions with existing anti-TB drugs. Fractional inhibitory concentrations of the compounds and the current anti-TB drugs should be determined by checkerboard assay in future studies.

In conclusion, our study successfully identified Ebselen as the most desirable antibiotic, while ST-193 and ST-193 (hydrochloride) were proposed to be potential candidates of anti-virulence for the *M. tuberculosis* complex.

## 4. Materials and Methods

### 4.1. Bacterial Strains

The *Mycobacterium bovis* vaccine strain, bacille Calmette-Guérin (BCG)-1 (Russia) (*M. bovis* BCG), which was used for the screening of small compounds, was collected from Queen Mary Hospital, Hong Kong. The organism was revived using Middlebrook 7H9 broth and 7H10 agar including OADC (10%), glycerol (0.2% in 7H9 broth and 0.5% in 7H10 agar), sodium pyruvate (4.4 mg/mL), and pancreatic digest of casein (1mg/mL) at 37 °C for 14 days.

*M. tuberculosis* H37Rv and two drug-resistant clinical isolates of *M. tuberculosis,* namely HKU1462 (an MDR-TB strain co-resistant to isoniazid and rifampicin) and WC274 (an XDR-TB strain resistant to isoniazid, rifampicin, pyrazinamide, ethambutol, streptomycin, fluoroquinolones, and injectable aminoglycoside), collected from the same hospital, were used to determine the inhibitory and bactericidal activities against *M. tuberculosis*. The resuscitation of the archived clinical isolates was performed using Lowenstein–Jensen (LJ) medium. The detailed drug susceptibility pattern of the clinical isolates is shown in Appendix A.

### 4.2. Construction of Lux-Based Promoter-Reporter Plasmids and a Negative Control

The Mycobacterial reporter plasmid pMV306G13+Lux was obtained from Brian Robertson and Siouxsie Wiles (Addgene plasmid #26160, Figure 1A) [30], including a kanamycin-resistant gene, *lux*CDABE, and a promoter G13 located between the NotI and NcoI restriction enzyme sites. The promoters P_hsp_60, deriving from pSMT3, were inserted into pMV306 vectors to form the integrating vectors -pMV306hsp. Subsequently, the *lux* operon (*lux*CDABE) from pMU1* was cloned into pMV306hsp, generating plasmids of pMV306hsp+Lux. Finally, G13 of *M. marinum* replaced P_hsp_60 in plasmid pMV306hsp+Lux to form *lux*-based promoter-reporter plasmids pMV306G13+LuxAB+LuxCDE (pMV306G13+Lux) [30].

To construct a lux-based phoP promoter-reporter, pMV306G13+Lux and *phoP* promoters were digested with NcoI and NotI (New England Biolabs, Ipswich, MA, USA), followed by DNA purification (QIAGEN, Hilden, Germany) and ligation. The purified plasmids (100 ng per reaction) and the target promoters were incubated with T4 DNA ligase (New England BioLabs, USA) overnight at room temperature. The G13 promoter in pMV306G13+Lux was eventually replaced by the *phop* promoter (Figure 1B, Appendix A) to generate pMV306PhoP+Lux.

The negative control -pMV306Adaptor+Lux was similarly created by replacing the G13 promoter with an adaptor (5′-GGCCGCTTAGATCTTTC-3′-, a random sequence without the promoter activity). All plasmids used in this study are listed in Appendix A.

### 4.3. Transformation of Lux-Based Reporter Plasmids into M. bovis BCG

*Lux*-based promoter-reporter plasmids (pMV306PhoP+Lux) and negative control (pMV306Adaptor+Lux) were transformed into *M. bovis* BCG by electroporation. The sequences of the plasmid were confirmed by PCR-sequencing. The *lux* signal of transformed *M. bovis* BCG was validated by an IVIS Lumina imaging system (Perkin-Elmer, Shanghai, China).

### 4.4. Validating the Correlation of the Lux Signal of the Promoter-Reporter Screening Platform with Phop Gene Expression in M. bovis BCG

Ethoxzolamide (ETZ) was shown to inhibit the PhoPR regulon in *Mycobacterium tuberculosis* via binding to *phop* promoter regions [31]. In this study, *M. bovis* BCG transformed with pMV306PhoP+Lux were treated with 200 μg/mL ETZ (experimental group) and DMSO (control group) at 37 °C for 24 h. The optical density via absorbance at 600 nm (OD_600_) and *lux* signals from all samples were measured by a Benchmark Plus Microplate Spectrophotometer (BIO-RAD) and a VICTOR3 Multilabel Plate Reader (PerkinElmer), respectively. The OD_600_ of the culture medium was used to eliminate the background noise. The statistical effect size (Z-factor) was calculated as follows: Z = 1 − [(3SD of sample + 3SD of control)/(|mean of sample − mean of control|), where sample is the *lux* signals after treatment with ETZ, while control is the *lux* signals without drug treatment. A Z-factor between 0.5 and 1.0 indicates an excellent assay [32].

For the *LuxA* and *phoP* expression analysis, total RNA was extracted from the experimental and control group for quantitative reverse transcription PCR (RT-qPCR). The fold-changes in gene expression of *luxA* and *phoP* were calculated based on a housekeeping gene *recX* [33]. The primers used for RT-qPCR are listed in Appendix A.

### 4.5. Screening Experiments

*M. bovis* BCG transformed with pMV306PhoP+Lux and pMV306Adaptor+Lux (negative control) were treated with 320 antiviral small compounds (MedChemExpress, USA) and three first-line anti-TB drugs (ethambutol, isoniazid, and rifampicin) (Appendix A). The assay was carried out in a 96-well plate. Each compound (2 μL, 10 mM) was dispensed in 100 μL of *M. bovis* BCG suspension (OD_600_ ≈ 0.2) in triplicate and eventually had a concentration of 200 μΜ. The concentration of the antiviral compounds used to treat MTB was indicated by a previous study that applied 140 to 351µM of anti-virulence drugs (L-ascorbic acid and 2P-AC) to treat MTB [8]. To eliminate the bactericidal effects of the three anti-TB drugs, their final concentrations were adjusted to two-fold lower than their MICs to *M. bovis* BCG, which were 3.8 μg/mL (ethambutol), 0.08 μg/mL (isoniazid), and 0.4 μg/mL (rifampicin) [34]. These compounds were dissolved in DMSO or water. Meanwhile, 100 μL of BCG suspensions without treatment were involved as drug-free control. OD_600_ and *lux* were measured after 4 h of treatment. Those showing *lux* inhibition (CPS-photon count per second ≈ 0) were selected. The selected *M. bovis* BCG suspensions were washed with 7H9 medium twice to remove residual compounds and then were inoculated on drug-free 7H10 medium, followed by incubation at 37 °C for 14 days to determine their viabilities.

### 4.6. Minimal Inhibitory Concentrations (MICs) and Minimal Bactericidal Concentrations (MBCs) against M. tuberculosis Complex

Compounds showing bactericidal effects on *M. bovis* BCG after 14 days of incubation were selected for further investigation. The MICs and MBCs of them against *M. bovis* BCG, *M. tuberculosis* H37Rv, HKU14621 (MDR clinical isolate), and WC276 (XDR clinical isolate) were determined using the microbroth dilution method according to the Clinical and Laboratory Standards Institute (CLSI) [35]. The selected compounds were diluted from 400 μM to 1.5625 μM using the “two-fold serial dilution” method. The MIC and MBC experiments were conducted in triplicate.

### 4.7. RNA-Seq Transcriptome Analysis

For RNA-seq analysis, untransformed *M. bovis* BCG (i.e., with no plasmid) was treated for 4 h with the selected compounds at a concentration of 200 μM or two-fold lower than their respective MICs if they exhibited inhibitory or bactericidal activity in previous MIC and MBC experiments. Total RNA was extracted with ribosomal RNA depletion after a 4 h treatment, followed by a strand-specific library (the bacterial cDNA library) construction with the quality control process. Subsequently, RNA-Seq was performed using the HiSeq X Ten system. Raw reads from RNA-seq were checked by FastQC v0.11.9 and trimmed by TrimGalore v0.6.7. The filtered clean reads were mapped to the reference genome (*Mycobacterium bovis* BCG strain Russia 368, complete genome) by HISAT2 (version 2.2.1). StringTie and ASprofile were used for transcript assembly and quantification, respectively. The data quality is shown in Appendix A. Cufflinks software was then used to quantify the transcripts and gene expression levels using mapped reads’ positional information on the gene. The gene expression levels were calculated in the form of fragments per kilobase million (FPKM) by GenomicFeatures (R package) and are listed in Appendix A. Genes with undetected expression levels in all the samples were excluded in the following analysis. The calculated FPKM was used to perform a principal component analysis (PCA) by the R function pcromp. A three-dimensional figure with the top three principal components (PC1, PC2, and PC3) was generated by R. The differential mRNA expression was analysed by DESeq2 using Trinity v2.8.4 with default parameters [36]. Pairwise comparisons and a clustering analysis were performed using the Trinity v2.8.4 package (Biostars, New Taipei, Taiwan). Genes with at least a 2-fold change with adjusted *p*  <  0.05 were identified as differentially expressed genes (DEGs). A gene ontology (GO) enrichment analysis was performed by the GENEONCOLOGY and PATRIC tools [37,38,39,40], and the chord plot was created by GO plot 1.0.2. Meanwhile, the mechanisms of the drugs that could kill *M. tuberculosis* were predicted and illustrated via the Kyoto Encyclopedia of Genes (KEGG) pathway database. In brief, significantly downregulated genes in *M. bovis* BCG treated with antibiotics (logFC < −2 and FDR < 0.01 compared with the DMSO control sample) that probably are critical for *M. tuberculosis* metabolism were selected and predicted as potential drugs of *M. tuberculosis* elimination. No biological replicates were conducted for the RNA-Seq experiment.

### 4.8. Cell Viability Assay—LDH Assay

The THP-1 monocytic cell line (ATCC TIB-202) was cultured in triplicate at a density of 5 × 10^4^ cells per well of a 96-well plate containing RMI1640 medium (GIBCO, Waltham, MA, USA) supplemented with 5% (*v*/*v*) fetal bovine serum (GIBCO, USA) at 37 °C with 5% CO_2_. The cells were then treated with 10 μL of sterile water and 10 μL of the compound (concentrations ranging from 0.2 μM to 200 μM) to determine the spontaneous LDH activity (control group one) and compound-treated LDH activity, respectively, using a CyQUANT™ LDH Cytotoxicity Assay Kit (Thermo Fisher Scientific, Waltham, MA, USA). Additionally, cells of control group two were treated with nothing to measure LDH activity.

### 4.9. Statistical Analysis

Data were expressed as means ± SD, and *p* < 0.05 was significant as measured by two-sample *t*-tests for the promoter-reporter validation and antiviral compound screening using GraphPad Prism (GraphPad Inc., San Diego, CA, USA). A two-way ANOVA was performed in the *phoP* and *phoR* expression investigation.

## 5. Conclusions

In conclusion, our study successfully identified Ebselen as the most desirable antibiotic, while ST-193 and ST-193 (hydrochloride) were proposed to be potential anti-virulence agents for the *M. tuberculosis* complex.

## Figures and Tables

**Figure 1 antibiotics-11-00369-f001:**
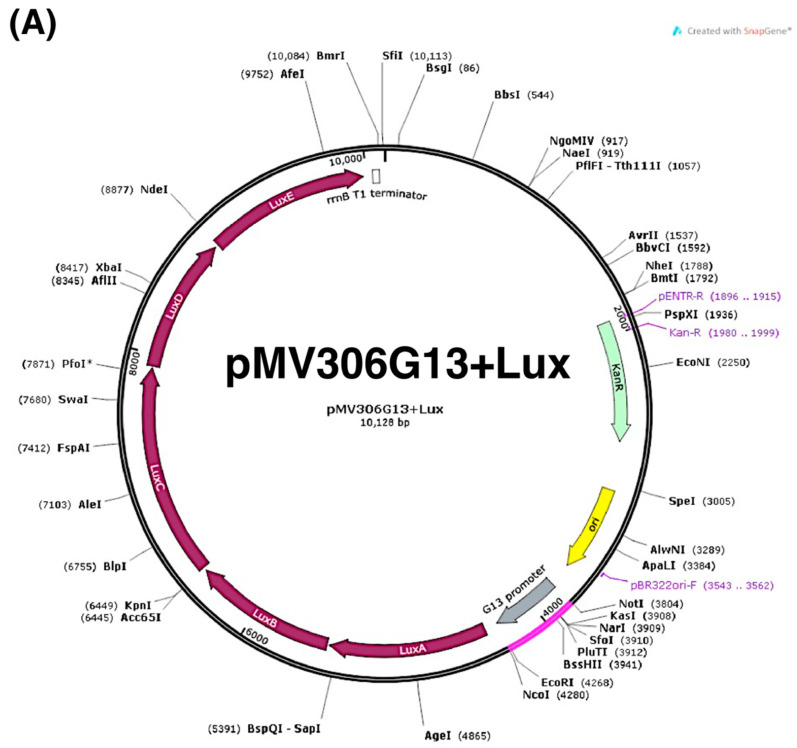
Construction and validation of the *lux*-based promoter-reporter platform. (**A**) pMV306G13+Lux. (**B**) pMV306PhoP+Lux. (**C**) Mean (± SD, *n* = 3) *lux* signal (in log lux) and OD_600_ in *M. bovis* BCG with and without (control) ETZ treatment at time 0 h and 24 h. (**D**) *LuxA* and *phoP* expression of *M. bovis* BCG with and without (control) ETZ treatment. Data are presented as means ± SD (*n* = 3) and analyzed using an unpaired *t*-test, *** *p* < 0.001 vs. BCG without ETZ treatment.

**Figure 2 antibiotics-11-00369-f002:**
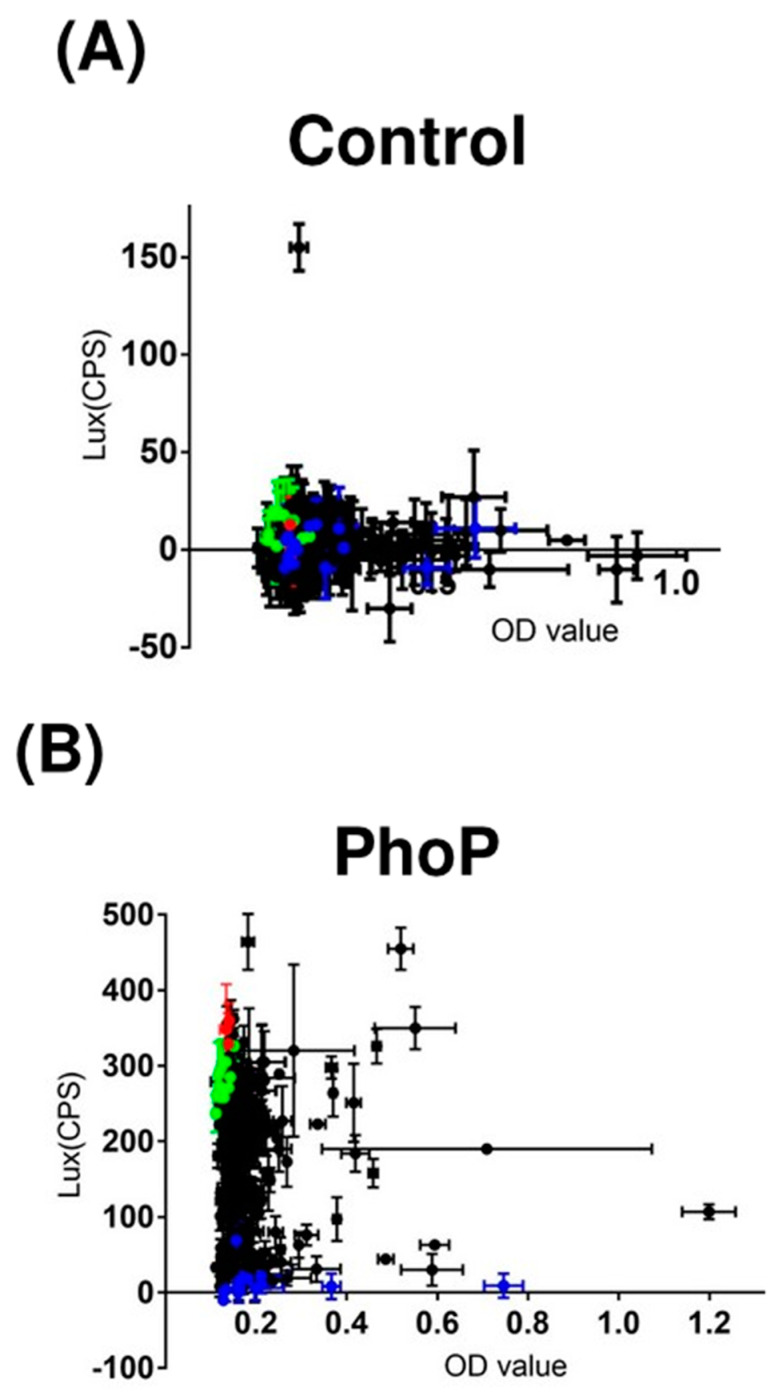
Screening of 320 antiviral compounds and 3 anti-TB drugs. (**A**) OD_600_ and LUX of all samples with/without 4 h compound treatment in negative control platform of *M. bovis* BCG. (**B**) OD_600_ and LUX of all samples in the *phoP* promoter-reporter platform. Green dots represent BCG samples (including pMV306Adaptor+Lux) without any compound treatment. Red dots represent samples treated with the three anti-TB drugs. Blue dots represent samples treated with the 16 selected compounds, while black dots represent samples treated with other compounds. (**C**) Comparison of lux signals between the compound treatment and control samples. ** *p* < 0.01.

**Figure 3 antibiotics-11-00369-f003:**
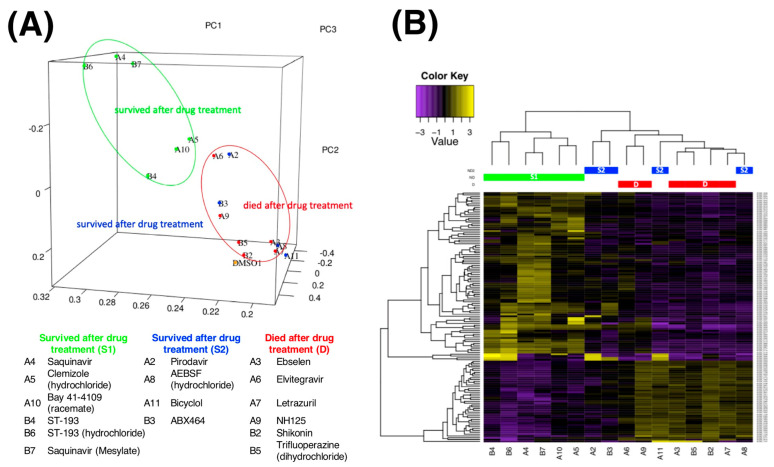
RNA-Seq transcriptome analysis and representative genes involved in the phoP- pathway. (**A**) Principal component analysis (PCA) of BCGs after treatment with 16 compounds in 3 groups based on the gene expression detected in all samples. (**B**) Heatmap showing a clustering analysis of gene expression levels for the three groups of BCGs (*p* < 0.05, FC > 2 or < 0.5). (**C**) Expression of *phoP* and *phoR* in S1, S2, D, and the control groups. Data are presented as means ± SD (*n* = 6 for S1 and D, *n* = 4 for S2, and *n* = 1 for the control group) and were analyzed using a two-way ANOVA, * *p* < 0.05, *** *p* < 0.001, while those without a label have no significances. (**D**) The expression of genes regulated by phoP/R in BCGs after treatment with the 16 compounds.

**Figure 4 antibiotics-11-00369-f004:**
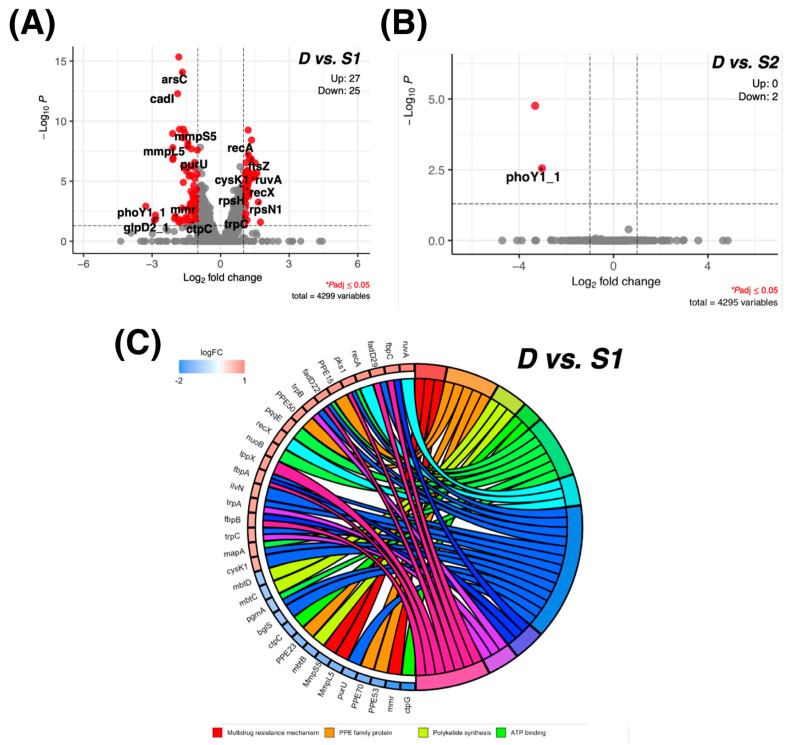
Molecular regulation associated with the anti-virulence process. (**A**) Volcano plot of the DEGs between D and S. (**B**) Volcano plot of the DEGs between D and S2. (**C**) Molecular functions and biological processes enriched in the DEGs for the survival groups.

**Figure 5 antibiotics-11-00369-f005:**
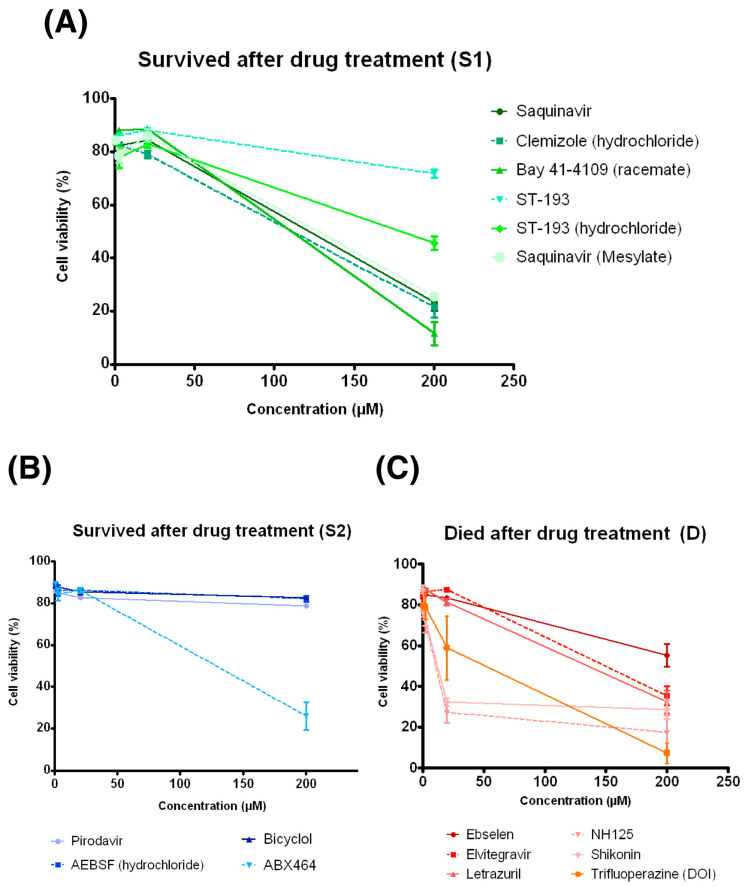
LDH results in 16 compounds in THP-1 cells. (**A**) Cell viability after being treated with drugs in the S1 group. (**B**) Cell viability after being treated with drugs in the S2 group. (**C**) Cell viability after being treated with drugs in the D group. Data are presented as means ± SD (*n* = 3).

**Table 1 antibiotics-11-00369-t001:** The MICs and MBCs of six compounds against BCG and MTB (*n* = 3).

	BCG		H37Rv		MDR-MTB		XDR-MTB	
Compounds	MIC (μM)	MBC (μM)	MIC (μM)	MBC (μM)	MIC (μM)	MBC (μM)	MIC (μM)	MBC (μM)
**Ebselen**	50	100	100	200	50	100	50	50
**Elvitegravir**	100	200	>400	>400	>400	>400	>400	>400
**Letrazuril**	25	100	200	>400	200	400	200	>400
**NH125**	25	50	50	100	25	50	25	50
**Shikonin**	25	50	50	100	25	100	25	100
**Trifluoperazine (dihydrochloride)**	12.5	25	25	50	25	50	25	50

## Data Availability

Raw data of the RNA-seq part from this study can be accessed through GSE190433.

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
