# Peer review of "Screening Repurposed Antiviral Small Molecules as Antimycobacterial Compounds by a Lux-Based phoP Promoter-Reporter Platform"

_antibiotics, 2022, doi:10.3390/antibiotics11030369_

Round 1
Reviewer 1 Report
The authors report a repurposing screen against M tuberculosis, in particular the ability to suppress the expression of the virulence gene phop.
In general the description of the results is very brief and superficial, e.g. “Mycobacterial reporter plasmid -pMV306G13+Lux and the lux-based phoP promoter-reporter plasmid 74 (pMV306PhoP+Lux) were constructed as shown in Figure 1A and 1B.” This does not help the reader not familiar with the system. This continues throughout the manuscript. Therefore, the authors are advised to clearly describe the methods and to throughly describe the results.
Minor comments:
Any reference to drug repurposing in the introduction is missing. Please add some recent reviews or success stories for M. tuberculosis.
The font size of the plasmid maps in Figure 1 too small and hardly readable. Please use a different approach to show.
Abbreviations should be explained the first time they are used (e.g. ETZ)
It is not clear in which format the screening assay was carried out, 96 well, or 384 well?
Usually, the Z’ value is reported for high-throughput screens in order to allow to assess the robustness of the assay. Please calculate it.
Please remove the “-” before the “μM” thorughout the manuscript, this is wrong
A description of the principal component analysis is lacking, software, parameters, etc
The interpretation of Figs 3A and 3B is
Fig 3C: The top line marked with * is not clearly positioned. Please check.
Caption for figure 5 should be included (n=?, ). I suggest to split the figure in according to the groups shown in Fig. 3A, also to improve readability.
Please check the whole text for spelling and grammar errors. Here are just a few examples.
l. 58 Over thousands of…
l. 195 have been suggested
l. 198: host cell members
l. 272 were shown
Reviewer 2 Report
In the manuscript, Zhu et al. presented a lux-based phoP promoter-reporter platform to screen for anti-TB drugs. They first demonstrated the lux signal is positively corelated with phoP expression level using ETZ. Later they tested 320 compounds using the lux reporter platform. They identified 16 compounds that gives low lux signal. Follow up the 16 compounds, the authors then did MIC and MBC test. In addition, they performed RNA-seq to try to identify the pathways and targets for the drugs. Overall, the manuscript is a good proof-of-principle paper for the lux-based screening platform. However, they study does have very limited novelty and significance.
Reviewer 3 Report
Tuberculosis (TB) continues to be a global public health crisis, with many new cases of the disease manifesting in drug-resistant or virulent forms. Research to find effective new drug therapies for TB has been greatly impacted by the Covid pandemic, and there is general agreement in the TB research community that some two years has been lost in the general drug discovery effort. Researchers are investigating novel chemical entities for TB treatment, re-investigating as lead compounds old TB drugs that had been bypassed decades ago for more effective ones, and they are also exploring the potential to use drugs in TB treatment that were already known for other purposes.
This manuscript reports on attempts to discover anti-TB agents from a library of known small-molecule antiviral drugs via a rapid screening system. It is thus one example of efforts by the TB research community to repurpose known drugs as antituberculars. In the present manuscript, more than 300 small molecules were screened for their ability to suppress the expression of a key virulence gene, phop, of M. tuberculosis complex using luminescence (lux)-based promoter-reporter platforms. The minimum inhibitory and bactericidal concentrations were examined against M. tuberculosis, including drug-resistant forms, and the surrogate organism BCG. Cytotoxicities to human macrophages were studied. The new data are important, and the methodology described will be useful to other researchers. The manuscript will merit acceptance for publication following some minor revisions.
The authors will need some help with idiomatic English.
In lines 40-44 there is a need for more explanation of the contents of references 5 and 6. another couple of sentences are needed to make this discussion clear.
Table S5 contains very useful data and this table certainly belongs in the main narrative of the manuscript, not just in the Supplementary Materials. But what were the positive and negative controls in getting these data?
The authors are wise to note that in the future "Fractional inhibitory concentrations of the compounds and current anti-TB drugs should be determined by checkerboard assay in future studies." This will be crucial in looking for antagonistic or synergistic interactions. It is to be hoped that the authors will themselves carry out this work.
